# The Tomato *ddm1b* Mutant Shows Decreased Sensitivity to Heat Stress Accompanied by Transcriptional Alterations

**DOI:** 10.3390/genes12091337

**Published:** 2021-08-28

**Authors:** Prashant Kumar Singh, Golan Miller, Adi Faigenboim, Michal Lieberman-Lazarovich

**Affiliations:** Institute of Plant Sciences, Agricultural Research Organization, Volcani Center, Rishon LeZion 7505101, Israel; prashantbotbhu@gmail.com (P.K.S.); golanm@volcani.agri.gov.il (G.M.); adif@volcani.agri.gov.il (A.F.)

**Keywords:** thermotolerance, high temperature, DNA methylation, crop productivity

## Abstract

Heat stress is a major environmental factor limiting crop productivity, thus presenting a food security challenge. Various approaches are taken in an effort to develop crop species with enhanced tolerance to heat stress conditions. Since epigenetic mechanisms were shown to play a regulatory role in mediating plants’ responses to their environment, we investigated the role of DNA methylation in response to heat stress in tomato (*Solanum lycopersicum*), an important vegetable crop. To meet this aim, we tested a DNA methylation-deficient tomato mutant, *Slddm1b*. In this short communication paper, we report phenotypic and transcriptomic preliminary findings, implying that the tomato *ddm1b* mutant is significantly less sensitive to heat stress compared with the background tomato line, M82. Under conditions of heat stress, this mutant line presented higher fruit set and seed set rates, as well as a higher survival rate at the seedling stage. On the transcriptional level, we observed differences in the expression of heat stress-related genes, suggesting an altered response of the *ddm1b* mutant to this stress. Following these preliminary results, further research would shed light on the specific genes that may contribute to the observed thermotolerance of *ddm1b* and their possibly altered DNA methylation status.

## 1. Introduction

The global rise in temperatures, particularly in crop production areas, poses a major challenge to yield quantity, quality and food supply. As ambient temperatures continue to change and extreme episodes occur more frequently, the ability of plants to cope with heat stress conditions is of great importance. When heat stress occurs during the reproductive phase of plant development, the observed consequences include morphological alterations of anthers, bud abscission and reduced fruit and seed set. Tomato (*Solanum lycopersicum*), an important vegetable crop worldwide, is very sensitive to heat stress. Tomato fruit set is optimal when the average day and night temperatures are in the range between 21 and −29 °C during daytime and 18 and −21 °C at night [1]. Heat stress conditions are generated when day temperatures exceed 32 °C, and increased in severity if night temperatures increase beyond 20 °C. When occurring for a prolonged period (Moderate Chronic Heat Stress—MCHS) during flowering and fruit development, tomato plants will show reduced fruit set, fruit weight, total yield and seed production [2,3,4,5]. Among the various approaches taken to mitigate the sensitivity of plants to heat stress, epigenetic research provides an innovative way to understand and utilize the complex interaction of plants with their environment. Several studies demonstrated the link between various epigenetic elements such as histone modifications, 5-cytosine DNA methylation, and the response of plants to heat stress [6]. For example, heat stress was proposed to inhibit lateral root primordium formation in maize by histone acetylation-mediated transcriptional regulation [7]. In rice, heat stress applied to flowering plants postfertilization led to a reduction in seed size and aberrant endosperm cellularization, in parallel to a sharp decrease in DNA methylation and H3K9me2 histone modification in the promoter of *OsFIE1* [8]. Silencing of the *MSH1* gene in tomato using RNAi resulted in increased fruit weight and number under high-temperature field conditions. These phenotypic changes were linked to DNA methylation as the methylation inhibitor 5-Azacytidine (5-AzaC) repressed the observed phenotypes [9]. 

DNA methylation is dependent on several well-known enzymes, including DECREASE IN DNA METHYLATION1 (DDM1). DDM1 is a SWI2/SNF2 chromatin remodeling protein that allows DNA methyltransferases to access heterochromatin thereby facilitating DNA methylation [10]. The role of DDM1 in transposable elements silencing through DNA methylation deposition was well established in *Arabidopsis thaliana* [10,11] and demonstrated using mutant lines also in maize, rice and tomato [12,13,14]. Interestingly, a recent publication provides evidence for the role of DDM1 in the response of plants to environmental conditions. RNAi-*DDM1* lines of poplar (*Populus tremula* × *Populus alba*) were more tolerant to drought-induced cavitation and presented thousands of differentially methylated regions along with transposable elements activation [15]. We aimed to assess the possible involvement of DDM1 in heat stress response in tomato. Tomato encodes two DDM1 genes, *SlDDM1a* and *SlDDM1b*. The *Slddm1aSlddm1b* double mutant plants display severe pleiotropic vegetative and reproductive phenotypes, whereas the single mutant lines display normal phenotype under optimal growth conditions [14]. In this brief communication paper, we present preliminary data showing improved thermotolerance of the *ddm1b* mutant and provide an initial demonstration of a unique transcriptional response under heat stress conditions. These findings may contribute to the understanding of the involvement of chromatin remodeling and DNA methylation in the response of plants to heat stress and may thus assist in future efforts to maintain crops yield in a warming climate. 

## 2. Materials and Methods

### 2.1. Plant Material and Growth Conditions

Seeds of the tomato (*Solanum lycopersicum*) *Slddm1b* mutant [14] of the T3 generationwere kindly provided by Dr. Tzahi Arazi (ARO, Volcani center, Rishon LeZion, Israel). *Slddm1b* T3 plants and the background cultivar M82 were grown in climate-controlled greenhouses at the Naan site of the Evogene LTD Company. Seven plants from each line were grown under either moderate chronic heat stress (MCHS) conditions (32 °C/22 °C day/night, starting at flowering) or control conditions (25 °C/18 °C day/night), in a randomized setup, identical in the two adjacent greenhouses. 

### 2.2. Reproductive Traits Measurements 

The number of flowers per inflorescence (FlN), fruit number per inflorescence (FrN) and fruit set rate (FS) were evaluated from 3–5 randomly selected inflorescences from seven different plants (a total of 25–35 inflorescences per line/condition). FS was calculated as follows: (FrN)/(FlN) × 100 = fruit set rate. Seeded fruit set ratio (SFS) was evaluated from 15–23 fruits, collected from seven plants per line/condition. SFS was calculated as follows: (No. of fully seeded fruits)/(Total no. of fruit sampled) × 100 = SFS rate. Seeded fruit was considered as such if more than ten seeds were observed in a transverse section. Seed number per fruit (SN) was examined by seeds extraction using all fruits (7–44) from all seven plants of each line and condition. Seeds were extracted using sulfuric acid as follows: the locular gel containing the seeds was extracted and soaked in a 2% sulfuric acid solution. After three hours, the seeds were transferred into a net bag and rinsed with tap water. Seeds were then thoroughly dried in open air for a few days. SN was calculated by weighing a small portion that was counted manually. Then the total amount of seeds was estimated by weighing.

### 2.3. Seedling Heat Stress Survival Assay

Seeds of the *Slddm1b* mutant and the background cultivar M82 were sown in cups filled with potting soil (10–13 seedlings per cup, 4–6 cups per line). The sowing date was adjusted up to two days per line to obtain seedlings of similar developmental stages. First, seedlings were grown in a controlled growth room (24 °C, 12 h light) for 10 days. Then, uniformly developed seedlings (just at the emergence of the first true leaf) were subjected to a heat stress treatment of 45 °C for 4 h. After recovery at control conditions (24 °C, 12 h light) for 1 week, seedlings were scored for survival rate. Seedling survival rate (SSR) was calculated as follows: (Total seedlings − dead seedlings)/(Total seedlings) × 100 = Survival rate. The entire experiment was repeated three times.

### 2.4. RNA Extraction, Library Preparation and Ilumina Sequencing

RNA was extracted from young leaves of the *Slddm1b* and M82 lines using RiboExTM (GeneAll; Seoul, Korea) according to the manufacturer’s protocol. Five to six biological replicates were used per line/conditions for a total of 22 samples.

RNA-seq libraries were prepared at the Crown Genomics Institute of the Nancy and Stephen Grand Israel National Center for Personalized Medicine, Weizmann Institute of Science. A bulk adaptation of the MARS-Seq protocol [16,17] was used to generate RNA-Seq libraries for expression profiling of the 22 samples. Briefly, (30 ng of input) RNA from each sample was barcoded during reverse transcription and pooled. Following Agencourct Ampure XP beads cleanup (Beckman Coulter, California, USA), the pooled samples underwent second strand synthesis and were linearly amplified by T7 in vitro transcription. The resulting RNA was fragmented and converted into a sequencing-ready library by tagging the samples with Illumina sequences during ligation, RT, and PCR. Libraries were quantified by Qubit and TapeStation as well as by qPCR for a 40S ribosomal reference gene (Solyc12g096540) as previously described [16,17]. Sequencing was carried out on a Nova-Seq6000 using SP, 100 cycles kit mode allocating 800 M reads in total (Illumina). The output was ~29 million reads per sample.

### 2.5. Bioinformatic Analysis

Poly-A/T stretches and Illumina adapters were trimmed from the reads using cutadapt [18]; resulting reads shorter than 30 bp were discarded. The remaining reads were mapped onto the *Solanum lycopersicum* genome, SL4.00, according to Sol Genomics annotations, ITAG4.0, using STAR [19]. Deduplication was carried out by flagging all reads that were mapped to the same gene and had the same UMI. Counts for each gene were quantified using the ESAT toolkit (ESAT. Available online: https://github.com/garber-lab/ESAT; accessed on 27 August 2021), using the BED file based on ITAG4.0 and the 3′ UTR region (1000 bases), and corrected for UMI saturation. Differentially expressed genes were identified using the DESeq2 [20] R package. The WGCNA R package was used for weighted correlation network analysis (https://horvath.genetics.ucla.edu/html/CoexpressionNetwork/Rpackages/WGCNA/; accessed on 27 August 2021). Gene ontology enrichment analysis was carried out using Blast2GO version 4.0 [21] based on Fisher’s Exact Test.

### 2.6. Statistical Analysis

The Students’ *t*-test was employed to identify significant differences (*p* < 0.05). All statistical analyses were performed using JMP Version 3.2.2 (SAS Institute, Inc., Cary, NC, USA).

## 3. Results

### 3.1. The ddm1b Mutant Is Less Sensitive to Heat Stress at Seedling Stage and Reproductive Development

In order to evaluate the response of the tomato *ddm1b* mutant to heat stress conditions, we first tested the recovery rate of seedlings from an acute heat stress treatment. In this experiment, 10-day old seedlings were exposed to a heat stress treatment of 45 °C for 4 h, followed by a recovery period of 10 days at 24 °C. Seedlings kept at 24 °C as a control group always survived at a 100% rate, whereas heat stress treatment led to 19% of the seedlings surviving in the M82 sensitive line. In the *ddm1b* mutant line, 36% of the seedlings survived the heat stress treatment (Figure 1a, Appendix A).

Following the finding that *ddm1b* is more heat stress tolerant than M82 at the seedling stage, we tested reproductive performance under heat stress conditions. To this end, we generated conditions of moderate chronic heat stress (MCHS) in a controlled greenhouse, allowing flowering, fruit, and seed set to occur under stressful conditions. Plants of *ddm1b* and M82 lines were grown under either stress (32 °C/22 °C day/night, starting at flowering), or control (25 °C/18 °C day/night) conditions, and scored for several reproductive traits, such as: number of flowers per inflorescence (FlN), number of fruits per inflorescence (FrN), fruit set ratio (FS), seeded fruit set ratio (SFS) and number of seeds per fruit (SN). The heat-sensitive M82 line presented a decrease in the number of fruits produced per inflorescence when grown under MCHS conditions, reaching an average of 2.8 fruits compared with five fruits under control conditions. The *ddm1b* mutant produced on average 4.3 fruits per inflorescence under MCHS conditions, which is significantly higher than the M82 line (Figure 1b, Appendix A). As expected, we found no difference in the number of flowers between control and MCHS conditions in both lines (Figure 1c, Appendix A). Nonetheless, this result shows that the improved production of the mutant under either condition was not due to changes in flowers number. The reduced fruit production, along with no change in flowers number (Figure 1c, Appendix A), led to an overall decrease in fruit set rate under MCHS conditions compared with control for both lines. Nonetheless, the *ddm1b* mutant reached a 56% fruit set which was significantly higher than M82 (37%, Figure 1d, Appendix A). In addition to fruit set, seeds production is also a major trait affected by heat stress conditions during flowering and fruit development. When we tested the number of seeds produced per fruit, there was indeed a reduction in MCHS versus control conditions in both *ddm1b* and M82 lines (Figure 1e, Appendix A). However, the rate of seeded fruits (fruits producing more than 10 fully developed seeds) in the MCHS greenhouse was dramatically higher in the *ddm1b* mutant compared with M82 (75% and 28%, respectively). Moreover, unlike M82, the rate of seeded fruits in *ddm1b* was not reduced by the stress conditions, and remained similar between control (91%) and MCHS (75%), (Figure 1f, Appendix A).

### 3.2. Transcriptomic Analysis Reveals Differential Response to Heat Stress between the ddm1b Mutant and M82

Following the observed reproductive thermotolerance of the *ddm1b* mutant, we aimed to characterize the response to heat stress at the molecular level. As a preliminary approach, we focused of leaf transcriptome, to reveal changes in gene expression that are unique to the heat stress response on *ddm1b*.

Using the Illumina platform, 22 cDNA libraries (five and six biological replicates per conditions for M82 and *ddm1b* lines, respectively) were sequenced. A total of ~704 M reads were generated, with an average of ~32 M reads per library. All reads were mapped to the tomato genome SL4.00. The average rate of gene annotation per library was 80.08%.

Using weighted gene co-expression network analyses (WGCNA), hierarchical clustering identified 31 modules of genes with similar expression patterns (Figure 2a,b). Furthermore, the module-trait relationship analysis identified gene modules linked with the stress condition, confirming that indeed, plants of both lines, *ddm1b* and M82, responded to the MCHS conditions applied, at the transcriptional level. Importantly, this analysis also identified gene modules that differ between the lines (e.x. MEsteelblue, MEgrey60 and MEroyalblue, Figure 2b demonstrating transcriptional alterations in *ddm1b* compared with M82).

Differentially expressed genes (DEGs) were identified with the DESeq2 program, using thresholds of *p*-value < 0.05 and |log2 (FoldChange)| > 1. Our results indicate that, under MCHS conditions, 504 and 385 transcripts were upregulated in *ddm1b* and M82, respectively (Figure 3a). Comparing the genes in these two groups, we found that only 55 genes (6.6%) were common (Figure 3b). Similarly, 368 and 387 transcripts were respectively down-regulated in *ddm1b* and M82 under MCHS conditions, with 47 (6.6%) common transcripts (Figure 3a,b). This result implies a different transcriptional response to heat stress in the *ddm1b* mutant. In addition, 489 transcripts were up- or down-regulated in *ddm1b* versus M82 under heat stress conditions (Figure 3a).

GO term enrichment analysis of the two groups of up-regulated transcripts under heat stress conditions confirmed the transcriptional response to heat stress, as both *ddm1b* and M82 transcripts were enriched in heat stress-related GO annotations. However, the number of genes within each GO group differed between the lines, mostly higher in *ddm1b* (Figure 3b). The MCHS down-regulated transcripts (368 in *ddm1b* and 387 in M82, 47 in common), were not enriched with a particular biological process.

Overall, we found that the *ddm1b* mutant showed improved heat stress tolerance both at the vegetative as well as the reproductive developmental stages, compared with the background cultivar M82. Preliminary analysis of the transcriptomic data confirmed that both *ddm1b* and M82 activated their heat stress response processes; however, the transcripts induced overall were largely different between the lines, suggesting a unique response of the *ddm1b* mutant that may be related to its thermotolerance.

## 4. Discussion

Plant productivity under environmental stress is a highly complex process, integrating multiple physiological traits and molecular pathways. In tomato, fruit production under heat stress conditions results in reduced fruit set which translates to yield loss [22]. In order to cope with this environmental sensitivity and allow normal fruit production under heat stress conditions, we need to understand the regulatory network that underlies the stress response. Epigenetic pathways, such as histone tail modifications and DNA methylation, are central in gene regulation in response to environmental cues [6]. In particular, DNA methylation was shown to contribute to increased fruit weight and number in tomato under high-temperature field conditions [9]. This reproductive vigor under stressful conditions was attributed to central components of the epigenetic machinery—METHYLTRANSFERASE 1 (MET1) and HISTONE DEACETYLASE 6 (HDA6), responsible for DNA methylation and histone tail deacetylation, respectively [23].

The DDM1 enzyme is a SWI2/SNF2 chromatin remodeling protein that allows DNA methyltransferases to access heterochromatin thereby facilitating DNA methylation [10]. Plants mutated in this gene exhibit loss of DNA methylation and activation of transposable elements [12,13,14]. Nonetheless, its role in plant stress tolerance was recently proposed. The DDM1-deficient line in poplar trees was shown to have improved drought tolerance accompanied by DNA methylation and transcriptional alterations [15].

We tested the response of the tomato *ddm1b* mutant to heat stress conditions, either by introducing acute heat stress at the seedling stage, or by generating moderate chronic heat stress (MCHS) conditions at the reproductive stage. In both heat stress regimes and developmental stages, *ddm1b* performed better than the M82 control background line. At the seedling stage, *ddm1b* presented a two-fold increase in survival rate, whereas at the reproductive stage, *ddm1b* produced higher quantities of fruits and seeded-fruits. Interestingly, this productivity advantage is not related to pollen development and performance, a central factor in the heat stress sensitivity of plants, as the number of seeds per fruit was reduced more significantly in the *ddm1b* mutant compared with M82, under MCHS conditions. Our results may serve as an important step in future development of heat stress-tolerant tomato cultivars, as well as other thermotolerant crop varieties. In order to gain insight into the genetic and molecular basis of *ddm1b’s* thermotolerance, we carried out a transcriptome analysis of young leaf tissue from plants grown either under control or MCHS conditions. This data set, albeit representing vegetative rather than reproductive tissue, provides a first glance into the transcriptional response of the plants to the imposed stress. Indeed, we could see that gene groups that are annotated to take part in response to heat stress were markedly induced, in both *ddm1b* and M82 lines. However, some differences were detected between *ddm1b* and M82 in regard to the heat stress related gene clusters, as expression patterns differ between the lines. In addition, 449 transcripts were upregulated uniquely in the mutant under stress; among these, three genes with GO annotation of ‘Hsp90 (chaperon) protein binding’ were induced only in *ddm1b*, but not in M82 (raw data presented in Appendix A) [24,25]. These initial indications point at a unique transcriptional response that possibly underlies the thermotolerance of the *ddm1b* mutant. Continuing analysis is being carried out to decipher the differences between the lines on a gene-specific level. As gene expression is highly dynamic, future experiments should focus on additional tissues and developmental stages, in particular those are directly linked to the observed increased fruit and seed set, i.e., inflorescence primordia and anther/pollen tissue at different stages of pollen development.

In the work presented here, we used *ddm1b* plants of the T3 generation. Since DNA methylation patterns may change in a transgenerational manner, the questions of phenotypic consistency along with transgenerational differences in DNA methylation patterns and gene expression are highly relevant and need to be tested.

Since we hypothesize that the phenotypic and transcriptional effects observed in *ddm1b* in response to heat stress are a result of its compromised methylome, our future goals also include analyzing the DNA methylation patterns in both *ddm1b* and M82 plants in response to heat stress conditions. Although, under non-stress conditions, global DNA methylation levels are only slightly affected in *ddm1b* [14], there might be specific changes that affects the expression of specific genes. Moreover, heat stress conditions may differently affect the methylome of *ddm1b* and M82, again leading to transcriptional differences.

Altogether, we have provided a first indication for improved heat stress tolerance of a *ddm1b* mutant in a crop plant, and created a preliminary data set to investigate the transcriptional heat stress response of this epigenetically perturbed tomato line.

## Figures and Tables

**Figure 1 genes-12-01337-f001:**
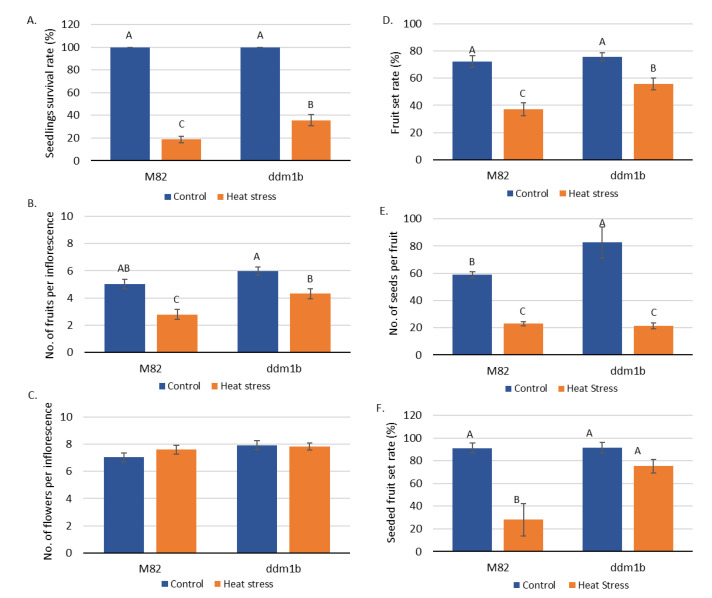
Phenotypic response of the *ddm1b* mutant to heat stress. Comparison between *ddm1b* and the M82 lines for seedlings survival (**A**), number of fruits per inflorescence (**B**), number of flowers per inflorescence (**C**), fruit set rate (**D**), number of seeds per fruit (**E**) and seeded fruit rate (**F**). Different letters denote statistically significant difference, *p*-value < 0.05.

**Figure 2 genes-12-01337-f002:**
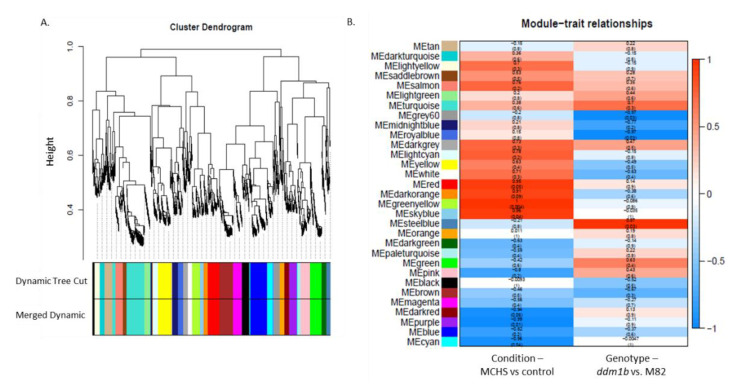
Expression profile-based hierarchical clustering. Using the WGCNA tool, a total of 31 distinct modules were identified. Gene dendogram obtained by hierarchical clustering with the module color (**A**) and relationships of modules and different samples (**B**) are presented. Each row in the table corresponds to a color module, and each column corresponds to a sample type.

**Figure 3 genes-12-01337-f003:**
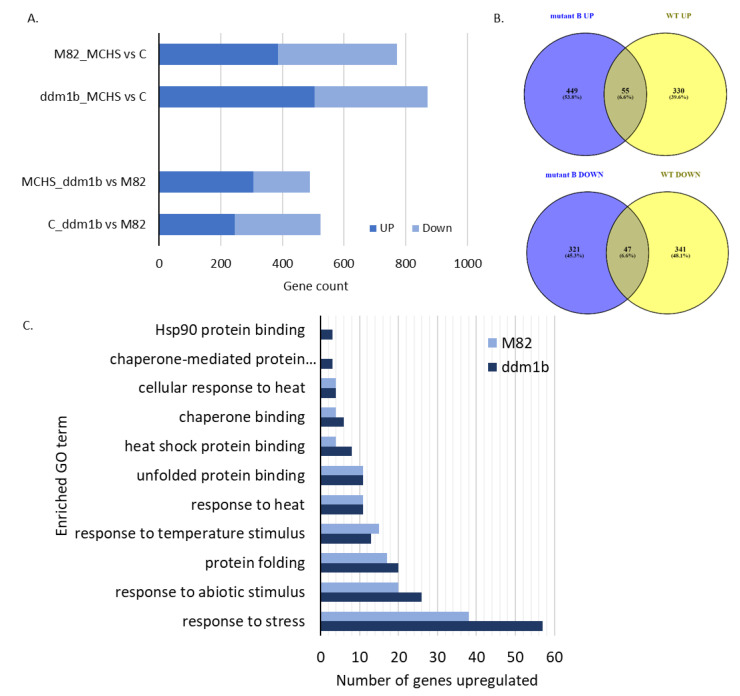
Heat stress-induced differentially expressed genes (DEGs) in *ddm1b* and M82. Transcriptome analysis results: number of DEGs in each group of line and condition (**A**), comparative view of DEGs between *ddm1b* (mutant) and M82 (WT) lines by up- and down-regulated genes (**B**), and the number of heat stress related genes upregulated in heat stress vs. control by GO biological process annotation (**C**). MCHS, moderate chronic heat stress conditions. C, control conditions.

## Data Availability

The data of this project have been deposited with links to BioProject accession number PRJNA750630 in The National Center for Biotechnology Information (NCBI) with BioSamples accessions: SAMN20500868, SAMN20500870, SAMN20500871 and SAMN20500872 for M82 ctrl, M82 heat stress, *ddm1b* ctrl and *ddm1b* heat stress, respectively.

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
