# Peer review of "The Tomato ddm1b Mutant Shows Decreased Sensitivity to Heat Stress Accompanied by Transcriptional Alterations"

_genes, 2021, doi:10.3390/genes12091337_

Round 1

Reviewer 1 Report

This paper describes the decreased sensitivity to heat stress accompanied by transcriptional alterations in the tomato ddm1b mutant using RNA-seq. The theme of this paper appears attractive and timely. The manuscript is simple and looks well written. However it still needs to contain some flaws. They used ddm1b mutant provided by other group, who made it by genome editing using CRISPR/Cas9. There is no description on which generation of the ddm1b seeds used in their experiments. How about the possibility of the different methylation levels as well as different expression levels of endogenous genes among progeny? What happens in the next generation in terms of consistency? These need to be considered in the manuscript. In Figure 1, they showed no significant difference between the control and ddm1 mutant under the conditions of heat stress in number of seeds per fruit (C) and number of flowers per inflorescence (E). Regarding to C, this is not significantly related to heat stress. The step from flower to fruit could be more sensitively amenable to heat stress applied. Some explanation needs to be added here or in the discussion. Besides that considering the total number of seeds of plants of both lines under control conditions, it may well be that ddm1b mutant was much more sensitive to heat stress conditions instead of heat stress tolerance (E). They did not mention this. Also they did not mention any difference in inflorescence numbers between the two lines. How about sizes of fruits and seeds between the two lines and two temperature conditions? Describe more in detail these issues using regular or supplementary tables showing numbers of them each plant. More detailed raw data would be valuable in this context. There are no descriptions on quality difference such as fruit weights. It would be very valuable to mention these matters in the manuscript. From these points of view, this manuscript needs much improvement and is not accepted in the present form for this Journal. Other comments are described below.

  • Figure 1: Add supplementary raw data (tables) as mentioned above.
  • Figure 2: This is composed of three figs. Three figures should be described as A, B and C. Besides that as a general the letter itself is too small to understand for readers. Only one figure could be appropriate in Figure 1. Some of them might be transferred to supplementary Figures to keep the letter size large enough for the readers to check.
  • Line 300: There is no expression data on HSP 90 in the text. At least the data on some highly increased expression level of genes should be shown in the text or supplementary table, for example, such as the top ten genes selected. Then the readers could understand easily. Describe any references on HSP 90 regarding direct heat stress tolerance.
  • In all 4 figure legends, capital letters should replace the lower case letters (such as Fig. 1a) in all Fig.1, 2, 3 and 4.

Author Response

Response to Reviewer 1 Comments

Point 1: There is no description on which generation of the ddm1b seeds used in their experiments.

Response 1: the seeds of the ddm1b used in our experiments were of the T3 generation. This information was added in the materials and methods section and in the conclusions section.

Point 2: How about the possibility of the different methylation levels as well as different expression levels of endogenous genes among progeny?

Response 2: This is indeed a very relevant and interesting aspect, which we are now testing. However, results are expected in a few months from now, thus, unfortunately, are not included in this short commentary. However, the importance of this aspect was noted in the conclusions section.

Point 3: What happens in the next generation in terms of consistency?

Response 3: We have tested T3 plants in this work. In future experiments, we plan to test consistency in the following generations. This is now added to the conclusions section.

Point 4: In Figure 1, they showed no significant difference between the control and ddm1 mutant under the conditions of heat stress in number of seeds per fruit (C) and number of flowers per inflorescence (E). Regarding to C, this is not significantly related to heat stress. The step from flower to fruit could be more sensitively amenable to heat stress applied. Some explanation needs to be added here or in the discussion.

Response 4: True, the number of flowers is not significantly related to heat stress and fruit initiation and development are more susceptible to heat. This result was included to show that the better fruit production is not due to increased flowering (figures 1b vs 1c). Moreover, this result supports, in a way, the thermotolerance effect of the mutant. Presenting a non-heat-stress trait that is not changed. An explanation was indeed missing and now added in the results section, trying not to have too strong interpretations.

Point 5: Besides that considering the total number of seeds of plants of both lines under control conditions, it may well be that ddm1b mutant was much more sensitive to heat stress conditions instead of heat stress tolerance (E). They did not mention this.

Response 5: Correct, when comparing the ratio of control/heat stress for the number of seeds, then the mutant is more dramatically reduced. This implies that the thermotolerance of the ddm1b mutant is not related to pollen development, a known, central factor in heat stress sensitivity in plants. A comment was added in the discussion section.

Point 6: Also they did not mention any difference in inflorescence numbers between the two lines.

Response 6: true, as inflorescence numbers were not scored in this experiment. We did, however, measured plant height, and saw no difference between the lines. Under the assumption (and non-scored observation)  that internode length is similar, we can assume a similar number of inflorescences. However, we do not have direct evidence to include in the MS.

Point 7: How about sizes of fruits and seeds between the two lines and two temperature conditions? Describe more in detail these issues using regular or supplementary tables showing numbers of them each plant. More detailed raw data would be valuable in this context. There are no descriptions on quality difference such as fruit weights. It would be very valuable to mention these matters in the manuscript.

Response 7: although fruits and seeds sizes/weights may be affected by heat stress, these were not included in our former experiments presented in this short communication paper. Detailed raw data for the traits included in this MS are now added as supplementary information.

Point 8: Figure 1: Add supplementary raw data (tables) as mentioned above.

Response 8: Tables of raw data are now added as supplementary information Tables S1-S4.

Point 9: Figure 2: This is composed of three figs. Three figures should be described as A, B and C. Besides that as a general the letter itself is too small to understand for readers. Only one figure could be appropriate in Figure 1. Some of them might be transferred to supplementary Figures to keep the letter size large enough for the readers to check.

Response 9: Indeed, the two clustering diagrams are redundant. To fit all reviewer's comments (I assume "figure 1" is meant to be "figure 2" in the original comment), I have removed one of them and made the module-trait scheme bigger. Section letters were added.

Point 10: Line 300: There is no expression data on HSP 90 in the text. At least the data on some highly increased expression level of genes should be shown in the text or supplementary table, for example, such as the top ten genes selected. Then the readers could understand easily. Describe any references on HSP 90 regarding direct heat stress tolerance.

Response 10: here we meant the genes that had GO annotation of Hsp90-binding, not the Hsp90 itself. That was indeed unclear in the text and modified in the revised version. Moreover, in order to give expression data as an example, we now provide the raw read data of these three genes in supplementary table S5. Since numerous publications describe a direct link of Hsp90 to heat stress, I cited a couple of review papers.

Point 11: In all 4 figure legends, capital letters should replace the lower case letters (such as Fig. 1a) in all Fig.1, 2, 3 and 4.

Response 11: lower case letters replaced by upper case letters

Reviewer 2 Report

This is a very exciting study that I think will generate much interest.  Heat stress effects will increasingly impact production of many crops, including tomato, a major vegetable crop which is relatively vulnerable to excessuve heat during the reproductive phase.  The authors show that methylation deficient ddm1b mutants are more tolerant to acute heat stress at the seedling stage and mild chronic heat stress during reproductive phase compared to the background genotype M-82.  They further show that ddm1b mutants have wide scale changes in their transcriptome response to stress compared to WT. The results shed light on the genetic control of heat stress responses in tomato and confirm that DDM1b is a promising target for genetic improvement of the tomato crop.

The experimental design is sound, the results are convincing, and the paper is clear and concise.  I have only a few very minor edits to suggest:

-L30 awkward wording; change to  '...the ability of plants...'
-L37 even temperatures of 32C would be considered heat stress for most genotypes of tomato
-L272 'ques' is spelled 'cues'

Author Response

Response to Reviewer 2 Comments

Point 1: L30 awkward wording; change to  '...the ability of plants...'.
Response 1: wording was changed

Point 2: L37 even temperatures of 32C would be considered heat stress for most genotypes of tomato.

Response 2: Agree, modified in the text.

Point 3: L272 'ques' is spelled 'cues'

Response 3: changed in the text.